# Stress and Resilience Experiences during the Transition to Parenthood among Belgian Lesbian Mothers through Donor Insemination

**DOI:** 10.3390/ijerph20042800

**Published:** 2023-02-04

**Authors:** Salvatore D’Amore, Alexandre Maurisse, Alessio Gubello, Nicola Carone

**Affiliations:** 1Centre de Recherche de Psychologie du Développement, de la Famille et des Systèmes Humains (DéFaSy), Faculté des Sciences Psychologiques et de l’Education, Université Libre de Bruxelles, 1050 Brussels, Belgium; 2Department of Brain and Behavioral Sciences, University of Pavia, 27100 Pavia, Italy

**Keywords:** lesbian mothers, donor insemination, sexual minority parenting, parenting stress, resilience, stigmatization

## Abstract

The present research explored parenting, stress, and resilience experiences among 16 Belgian, lesbian, first-time parental couples with donor-conceived children aged 3–72 months. In each couple, both mothers participated in a conjoint, semi-structured interview focused on their parenthood desire; the impact of stigmatization and social support from families of origin, friends, and institutions; and couple and family resources. Interviews were audio recorded, transcribed, and analyzed using Braun and Clarke’s reflective thematic analysis. Four themes were identified: (1) “The precious baby”: Realizing the parenthood project; (2) “Can we show ourselves in public without prying eyes?”: Family social visibility; (3) “It’s complicated!”: Parental legal recognition and role imbalance; and (4) “How can we handle this?”: Family resilience. The themes indicated that the child’s donor conception, the parents’ coming out, the non-gestational mother’s role, the legal obstacles encountered, and the need to find a balance between the two mothers in childcare tasks generated stress and required mothers to develop resilience strategies. The results suggest several potential areas for mental health practitioners to explore in clinical contexts when supporting intended lesbian mothers in their transition to parenthood through donor insemination.

## 1. Introduction

Despite the increase in research on lesbian and gay parenthood [1,2,3,4,5], there remains a lack of knowledge about parenting experiences and life cycle development among lesbian and gay individuals following their transition to parenthood [6,7,8,9,10]. According to Carter and McGoldrick’s [11] family life cycle model, the transition from two to three in the family group is characterized by important emotional processes. Indeed, this theory frames general functional and dysfunctional relational patterns within the development of the family as a system moving through time. As these changes emerge, the family must be able to adapt accordingly in order to avoid dysfunction. This involves the resolution of developmental tasks specific for each stage of the life cycle. In this vein, the authors divide the life cycle of the family into six stages (i.e., leaving home, the formation of the couple, the transition to parenthood, the family with teenagers, the empty nest, and the family in late life). For each stage, second-order changes are identified as fundamental in order to proceed properly to the next stage.

The parental couple has to deal with the acceptance of a new member (i.e., the child) into the system and must redefine and re-adjust the dyad to form a triad. Further developmental family tasks include parental coordination in child-rearing, management of financial and household tasks, and the realignment of relations with extended family members (e.g., grandparents). Accordingly, the transition to parenthood marks a new identity for the romantic couple that affects not only the romantic relationship, but also the overlapping parental system [11]. Furthermore, as parenthood is socially valued, and considered desirable and positive by most adults, it is laden with social expectations, which new parents may experience as significant stressors [12]. Social representations of parenthood, parents’ personalities and family histories, and the child’s temperament are among the key determinants [13] of parents’ approaches to raising their child, adjusting to parenthood, and transmitting internal representational models to their child. The present study explored the parenting, stress, and resilience experiences of lesbian mothers through donor insemination.

### 1.1. The Transition to Parenthood among Lesbian Couples through Donor Insemination

Given societal changes in family formation and the proliferation of more inclusive paths to parenthood, according to Carter and McGoldrick [11], the family life cycle can no longer be seen as universal and rooted in the nuclear family (i.e., two cisgender heterosexual parents raising their spontaneously conceived child). Following the women’s liberation and gay rights movements of the 1970s, the introduction of in vitro fertilization (IVF) in 1978, and the initiation of more equitable access to adoption for couples with diverse sexual orientation, the number of lesbian households increased significantly, in what has been described as a “lesbian baby boom” [14].

Nonetheless, in 2017, the National Consultative Ethics Committee (CCNE) reported that only eight European countries had legalized donor insemination for lesbian couples (i.e., Belgium, Denmark, Finland, Spain, Portugal, Sweden, the Netherlands, and the United Kingdom). In all other countries, lesbian couples were forced to go abroad to conceive, facing medical, legal, and psychological difficulties with respect to access to medical centers, ethical and moral arguments against donor insemination, and homonegativity from practitioners [15]. In 2022, the ILGA map [16] reported that seven further countries allow donor insemination for lesbian couples (i.e., Austria, France, Iceland, Ireland, Luxembourg, Malta, and Norway). For these reasons, the journey to parenthood for lesbian couples requires significant reflection, negotiation, and financial investment [17].

The present study was conducted in Belgium, which was pioneering in its recognition of coparent adoption in 2006, aimed at reducing parenthood inequalities for sexual minority couples through donor insemination. According to the law on adoption among same-sex parents, the non-gestational mother in a female parented couple had to adopt her child in order for her to be legally recognized as a mother, a decision that legally established a disparity between different-gender couples who recurred to donor insemination and lesbian couples who went through the same procedure. In 2014, a new law was enacted, allowing from 2015 the lesbian non-gestational mother to be legally recognized even before the child’s birth, in force of a joint parenthood project with her partner. This law promotes a “dual filiation kinship” in female-parented couples who recur to donor insemination, in an effort to preserve the child’s interests. Since then, the recognition of the non-gestational mother in Belgium has become automatic in the case of the lesbian couple being legally married when the child is born; otherwise, a recognition practice has to be issued with the local authority even before the birth of her child or within the first year after birth [18]. Thus, lesbian mother families continue to be challenged by legal and economic obstacles. This supports Gross’s [19] idea that sexual minority parent families do not experience emotional, biological, and legal ties as being synonymous (i.e., the heteronormative model). It follows that any reinforcement of this model puts significant pressure on sexual minority couples and makes their transition to parenthood unequal to that of heterosexual couples.

In their research on lesbian mothers with young, donor-conceived children in the Netherlands, Bos and colleagues [20] found that the mothers’ experiences of parental stress were comparable to those of heterosexual parents. Of note, a recent qualitative study by Farr and Tornello [21] showed that lesbian, gay, and heterosexual couples had similar experiences when they became parents, and that, regardless of parents’ sexual orientation, the transition to parenthood brought about significant changes to the couple’s relationship. However, although comparative research agrees that the transition to parenthood requires the intended parents to cope with similar challenges and adjustments, evidence also shows that lesbian and gay couples are additionally faced with stigmatization [21,22,23].

If societal stigma is internalized by lesbian and gay intended parents, it can challenge or even delay their journey to parenthood. In a study by Fossoul and colleagues [15], half of the lesbian couples reported experiencing varying degrees of difficulty when deciding to become parents. Such difficulties included internalized homophobia, lack of social support, indecision with respect to the donor type (i.e., known, anonymous, open-identity) and the choice of (non-)gestational mother, and financial and psychological strains associated with the insemination procedure.

However, Bos and colleagues [20] showed that, despite experiencing societal and internalized stigma, Dutch lesbian mothers through donor insemination felt more competent and less overwhelmed than heterosexual parents through spontaneous conception. Furthermore, relative to heterosexual mothers, lesbian biological mothers reported that it was significantly less important to them that their child developed societally valued qualities (i.e., ambition, self-control). However, lesbian non-gestational mothers felt more pressure to prove their parenting quality, relative to heterosexual fathers. More recent cross-cultural research in France, the Netherlands, and the United Kingdom has indicated that lesbian mothers through donor insemination tend to experience more positive feelings during pregnancy and report higher levels of parenting skills 4 months after the birth of their child, compared to gay couples through surrogacy and heterosexual couples through IVF [10].

### 1.2. Stigma Generating Stress for Lesbian Mother Families

Sexual stigma refers to the negative regard, inferior status, and relative powerlessness that society accords to individuals associated with non-heterosexual behaviors, identities, relationships, and communities [24]. When a sexual minority parent internalizes such negative societal attitudes against homosexuality and (unconsciously) accepts these attitudes as part of their value system and self-concept (i.e., suffers from internalized sexual stigma), they may feel less able to raise their child well. Internalized sexual stigma also encourages heteronormativity, promoting the idea that only gender conventionality, heterosexuality, and family traditionalism are “correct” and appropriate for parenthood [25]. In a similar vein, lesbian women may struggle with confronting the heterosexist norms according to which lesbianism and motherhood are incompatible [26,27]. Thus, lesbian intended mothers who also suffer from internalized sexual stigma may experience significant discomfort and stress. According to Goldberg [28], internalized sexual stigma may even moderate the process of becoming a parent.

In this vein, Bos and Gartrell [1] highlighted that lesbian mothers typically cite raising a child in a heterosexist society as a significant challenge. In this context, they may experience rejection in the form of regular irritating and intrusive questions regarding their family life, and they may also doubt their own parenting competency, as also indicated by further recent studies (e.g., [27,29]). This may be reflected in the offspring’s experience. A study conducted by van Gelderen et al. [30] found that, when adolescents were interviewed about their experiences of being raised by lesbian mothers, they reported significant stigmatization from peers, resulting in exclusion, ridicule, or rejection, due to their mothers’ non-heterosexual orientation. In addition, lesbian mothers’ families of origin may be reluctant to accept their parenthood project, especially when they have not yet accepted their daughter’s identity as a lesbian woman [31]. Family validation is also important for the non-gestational mother’s role identity development and comfort [32].

Of further relevance is that, when lesbian couples overcome stigma and initiate their parenthood project, other stressors may emerge, relating to the psychological, temporal, and financial costs of donor insemination procedures, as well as the potentially multiple attempts needed for the intended gestational mother to conceive (due to the experiential burden of hormonal treatments, the complexity of the medical procedures, and the probable failure of these procedures). Such difficulties often result in sudden disillusionment, which intended mothers must manage and recover from, in order to persevere in their project [33]. Additionally, the non-gestational mother may suffer from invisibility and a lack of recognition in her maternal role, to the extent that only the gestational mother is legally recognized as a mother [32].

### 1.3. Strategies for Adapting to Minority Stress and Building Resilience

Coping strategies describe a set of cognitive and behavioral efforts aimed at controlling, reducing, or tolerating internal or external demands that threaten or exceed one’s resources [34]. Most authors define family resilience as the family’s capacities and coping strategies to ensure its own well-being, as well as the well-being of its members [35]. With regard to lesbian mother families, specifically, previous qualitative research has highlighted some adaptive strategies that can provide positive results for stress management in a heteronormative society.

#### 1.3.1. Family Balance

Some lesbian mother families use a number of adaptive strategies to maintain equity in their transition to parenthood, including choosing names for each mother and dividing work and parenting roles [28,36]. Indeed, when lesbian women become parents, they must determine the terms with which their children will address them. In some cases, this task may be tense or complex. As a strategy, lesbian mothers may collaboratively select parallel names—that is, parallel derivative forms of mother (e.g., Mommy and Mama), identical derivative forms distinguished by each parent’s first name or initial, or a derivative form for the gestational mother and a derivative of mother from another language or culture for the non-gestational mother [37]. Parallel terms of address are likely used out of a desire to convey equal parenting identities and to solidify the non-gestational mother’s parental identity, as well as to express the mothers’ great effort and desire to maintain balance between them. Once the child is born, lesbian mothers are more likely to adopt an equity-based structure, particularly with respect to childcare and household tasks, relative to heterosexual couples [38,39,40]. This more equal task sharing is likely to create a climate of balance in the parental roles, linked with greater satisfaction in the couple relationship, which is an important protective factor against stressors [21,28,41].

#### 1.3.2. Community Change

When lesbian couples become parents, their contact with the LGBTQ+ community (and particularly community members who are not parents) is likely to decrease [28]. In the United States National Longitudinal Lesbian Family Study (NLLFS), lesbian mothers’ social support networks included more parents than non-parents, and many heterosexual individuals [42]. It is reasonable to conjecture that lesbian mothers may socialize with other parents in order for their child to interact with other children; such socialization also expresses mothers’ greater focus on their parental identity than their lesbian identity (which is traditionally viewed as childfree) [43]. However, mothers who remain engaged with the LGBTQ+ community may experience benefits, to the extent that such engagement moderates the negative impact of homophobic stigmatization on their psychological adjustment and ensures that their children are exposed to other children of sexual minority parents, thereby cultivating positive feelings towards their family diversity [44].

#### 1.3.3. Confrontation (Visibility Strategy)

In a study on the strategies used by sexual minority parents to protect their children from stigmatization, Gross [45] found that lesbian mothers implemented an activist visibility approach to normalize their family structure and educate society about family diversity. Indeed, sexual minority families have been shown to talk about their family composition from the first visit to a preschool or school, while judging reactions. This strategy may help parents select their child’s school and reduce the risk that their child will suffer from homophobic remarks [28,45].

#### 1.3.4. Preparation for Heterosexism

Lesbian mothers have been found to make significant efforts to prepare their child for stigmatization and questions from peers, and to help their child internalize a sense of legitimization about their family composition [46,47,48]. Most of what is known about sexual minority parents’ socialization approaches to their child’s minority status derives from studies with adoptive lesbian and gay parent families. Such studies have shown that most parents take an engaged approach to socialization about their child’s family status, initiating parent–child conversations aimed at instilling pride, seeking communities that reflect their child’s identities, and educating their child about heterosexism [49,50,51]. However, some parents take a more cautious approach, acknowledging their child’s family status but taking care not to be overly focused on their points of difference [49]. Some parents also minimize negative incidents and experiences related to heterosexism, which are nevertheless obvious, in an attempt to decrease their significance and limit their negative impacts on the child [52].

## 2. Materials and Methods

The present study aimed at describing the experiences of lesbian mothers through donor insemination, focusing on their sources of stress and discrimination and their strategies of resilience.

### 2.1. Participants

A total of 16 Belgian, lesbian, first-time parental couples (gestational mother’s age M = 33.68 years; SD = 6.12; non-gestational mother’s age M = 35.30 years; SD = 6.12) with donor-conceived children (M = 25.30 months; SD = 21.30; age range: 3–72 months) participated. Thirteen couples were legally married, while the remaining three were unmarried. The inclusion criteria were: (a) identifying as a cisgender lesbian woman, (b) being in a lesbian couple, (c) having a child conceived through donor insemination, and (d) residing in Belgium. Participants were recruited through snowball sampling techniques (i.e., word-of-mouth and a research flyer announcement posted on the websites of LGBTQ+ associations and Facebook groups for sexual minority parents).

### 2.2. Procedure

The study was approved by the Ethics Committee of the Université Libre de Bruxelles. Mothers provided informed consent to participate and did not receive any compensation. In each parental couple, mothers took part in a joint semi-structured interview at home, between January 2017 and January 2019. The research questions were: When and how did you become a couple? When and how did you plan to become parents? How did you experience your transition to parenthood? How did your families of origin, friends, and colleagues react to your decision to become parents? Did you experience any sources of stress and support during your transition to parenthood? Interviews lasted, on average, 40–60 min, and were audio recorded and transcribed.

### 2.3. Data Analysis

Data analysis followed the principles of reflexive thematic analysis (TA)—a subtype of thematic analysis that emphasizes researcher subjectivity and reflexivity [53,54]. The first stage of reflexive TA involves data familiarization: all interviews were read, reread, and coded by a single coder with no coding framework in mind. The data and codes were then re-examined with the research questions in mind (i.e., lesbian mothers’ specific experiences of stress and resilience during their transition to parenthood). Themes focused on participants’ experiences, and were revised and reviewed throughout the analytic process, via a back-and-forth analysis of the interview transcripts, codes, and themes between three coders. Below, the data are presented verbatim, although certain repeated words and filler words (e.g., “like,” “you know”) were tidied up [55], in order to reduce the discrepancy between edited academic writing and unedited speech. Pseudonyms are used to protect participants’ identities.

## 3. Results

Most of the collected data refer to the lesbian mothers’ experiences during their transition to parenthood and the initial months following their child’s birth. Four themes were identified: (1) “The precious baby”: Realizing the parenthood project; (2) “Can we show ourselves in public without prying eyes?”: Family social visibility; (3) “It’s complicated!”: Parental legal recognition and role imbalance; and (4) “How can we handle this?”: Family resilience.

### 3.1. “The Precious Baby”: Realizing the Parenthood Project

The first theme, “The precious baby”, reflects childbirth as the realization of the mothers’ plans for and journey towards parenthood, including their identification of which partner would be the gestational parent, their transition from being a couple to becoming parents, and their reconnection with their families of origin. In the words of Manon (gestational mother of a 3-month-old son): “I have had the desire for a child since I knew that women could have them. It was inconceivable for me not to have a child”.

After the birth, the baby is at the center of the family and becomes a priority. “As our center of interest”, Constance (gestational mother of a 22-month-old daughter) described, “I don’t really have any part away from her. The little baby is all”. Louise (non-gestational mother of a 16-month-old daughter) added, “She knows very well that she is the center of interest and that everyone revolves around her. She understands that very well”. The theme may also suggest renewed ties with the mothers’ families of origin, following the couple’s reconciliation with parents and grandparents. As reported by Alicia (gestational mother of a 7-month-old son): “He’s the mascot, he’s the focal point of the family, he’s tightened everything up, he’s adored by everyone, but you’ll see he’s cute! This pride in having this child, I think that in addition to being an adorable child, it’s part of a symbol, even if I did not make a child for a symbol”.

### 3.2. “Can We Show Ourselves in Public without Prying Eyes?”: Family Social Visibility

The second theme pertains to negative judgment towards the lesbian mother family, in the form of microaggressions, marginalization, and a lack of understanding. Once their child was born, some mothers found it difficult to expose themselves in public, due to indiscretion from the general public. For instance, while walking, some mothers were questioned about which one of them was truly the baby’s mother. Their answer—some form of “Beh, both of us”—would often result in astonished looks. To avoid any awkward feeling of unease, some mothers would simply avoid confrontation or exposure, making efforts towards invisibility, instead. The approach of Héloïse (non-gestational mother of a 4-month-old son), “to live happily, let’s stay hidden”, seemed justifiable and was practiced by some couples.

Additionally, many of the mothers reported encountering gratuitous insults, often on social networks. Victoria (gestational mother of an 8-month-old-daughter) was dismayed when she discovered some heinous comments left in response to articles about sexual minority parenting: “I always wonder about these people who spend their lives answering on forums, because here, on the other hand, I have already read some very violent stuff! If there is an article on same-sex parenting, this is where you see the comments!”

As lesbian mother families are not yet fully recognized in the eyes of society, many of the mothers felt that they were not perceived as normal families, and this increased their internalized heteronormativity. “I did not choose at all to fall in love with Pascaline (non-gestational mother of 6-month-old twin sons)”, said Hélène (Pascaline’s gestational coparent). She continued: “To choose, I was with a boy, and I would have preferred to continue. It would have been much easier. Afterwards, I have no regrets, but I want to say that it would have been easier. Now, we know that we are not the norm. We know that it challenges”.

### 3.3. “It’s Complicated!”: Parental Legal Recognition and Role Imbalance

The third theme refers to the mothers’ lack of legal recognition of parental rights prior to the change in law recognizing the dual filiation kinship for children born in lesbian mother families through donor insemination. In the previous legal context, the non-gestational mother, when unmarried and not the adoptive parent, had no rights and no obligations towards the child, leading to fear over a loss of parental rights. As reported by Elodie (gestational mother of a 1-year-old daughter): “Alice (Elodie’s non-gestational coparent) had absolutely no rights at all. From the moment the child was born, she had nothing more to say. It was quite complicated for us, and it worried me a bit. There was a long journey for my partner to legally become a mother to her daughter”.

The mothers wanted to live as two mothers, without hierarchy or distinction, and with equal parental rights. They did not want to be ousted from their family, confronted with the judgments of others, discriminated against, or subjected to endless administrative red tape. They also wished to avoid legal obligations to adopt the child from the “other mother”, in order to be officially recognized in the eyes of the law and others. Likewise, they did not want to feel apart from and left out of their family. However, the non-gestational mother did not have the same legal status in parental recognition compared to the gestational mother, and for many of the non-gestational mothers, fear, waiting, stress, and apprehension manifested in an emotional roller coaster. “The most stressful thing”, said Marie (non-gestational mother of a 4-year-old daughter), “was learning from the administrative employee that I was not her mother!” Pauline (gestational mother of a 3-year-old son) likewise reported, “But that too, anxiety. We went to the town to get the identity card. I saw my wife whitening in fear that they would tell us that I was the mom and she was no longer the mom!”

Some of the mothers remembered feeling non-existent or unrecognized by authorities during simple procedures such as the registration of their child with the municipality. One can only imagine the heartbreaking feeling that Alice (non-gestational mother of a 1-year-old daughter) must have experienced when, at one of these appointments, the clerk responsible for the administrative task spoke only to the woman who was registered as the “mother” of the child: “They only spoke to you because I didn’t exist”, said Alice.

This concern extended even further when the mothers considered the risk that the gestational mother could die and the child could be taken away from the other (i.e., non-gestational) mother and placed with another family. Such thinking could generate intense anxiety for both mothers. Indeed, Valentine (non-gestational mother of a 4-year-old son) raised the fact that even being in a civil partnership “would not change anything” and concluded that “it is an aberration”. For some mothers, this phase of administrative and legal procedures surrounding the non-genetic mother’s adoption of the child was experienced as “the only, the real difficulty”.

### 3.4. “How Can We Handle This?”: Family Resilience

The fourth theme illustrates anti-stigmatization strategies and resources to increase family resilience by seeking social and family support and preparing for potential insults. Commonly, the mothers used humor and self-deprecation as weapons to evade difficult and unpleasant social situations and reinforce the couple’s complicity, as explained by Pascaline (non-gestational mother of 6-month-old twin sons), who used humor to defuse vulgar remarks. In a similar vein, Hélène (Pascaline’s gestational coparent) ironically reported that, while filling in administrative papers, she retorted to the employee who asked her the name of the child’s father: “Brad Pitt!”

With the aim of strengthening couple cohesion and harmony, the mothers widely practiced task sharing. As Mila (gestational mother of a 23-month-old daughter) explained, “We always do everything together”. Likewise, Elodie (gestational mother of a 1-year-old daughter) insisted that she could count on her partner, under all circumstances, to avoid feeling overwhelmed: “If all is not ready, it does not matter, because Alice manages”. A balanced sharing of tasks and parental roles was very common within these couples, constituting a powerful internal family resource. In Lea’s (non-gestational mother of a 2-year-old daughter) words, “There is one who is going to get Camille up and dress her, while the other goes down to prepare lunch. She does the housework while I go to bathe”. Likewise, Jennifer (gestational mother of a 7-month-old daughter) reported, “Me, I choose her clothes because Adèle can’t match”. Adele (Jennifer’s non-gestational coparent) added: “Jennifer was breastfeeding her, so I was the one giving her the bath to have something intimate, just the two of us. And if not, well we are really interchangeable”.

An additional heavy and significant factor for the mothers was criticism about the lack of a male figure, as they did not want their child to suffer from a lack of a father. To overcome this, some mothers (especially those who had used an anonymous donor) wanted their child to be surrounded by male individuals. Adrianne (gestational mother of a 2-year-old daughter) explained that she and her coparent wanted a man (e.g., a brother, the designated godfather, a friend) who was close to their family circle to play a significant role in their child’s life. They believed that, as long as he was a good influence on the child, and was involved during the child’s development, they would make him an integral part of the family.

Another aspect raised by the mothers related to their terms of address. In this regard, Hélène (gestational mother of 6-month-old twin sons) explained that she did not hesitate to show her children photos that clarified who she and her partner were: “That’s mum and mum, because you have two mums”. Pauline (gestational mother of a 3-year-old son) and her partner opted for “mum” and “mamour”, while Pascaline and Hélène (non-gestational and gestational mothers of 6-month-old twin sons, respectively) preferred to use “mama” and “babou”. Some mothers used more normative terms, such as “the two mothers” or “mum” and “mum”. In the mothers’ view, all of these strategies helped their child(ren) decide what to call them.

Such naming strategies also served to educate teachers and other families at school. However, the mothers were also particularly attentive to the choice of their child’s school, a decision that, in some cases, was made even before the child’s birth. The mothers felt that openness and communication were important principles to pass on. Their careful consideration of schools related to their awareness that many difficulties could be experienced at school, such as stigmatization, micro-aggressions, and harassment. Thus, many mothers directly questioned and confronted school principals before selecting a school. As Pauline (gestational mother of a 3-year-old son) described, “Whether it’s at school, in relation to teachers and all that, well, when we talk about it, they know that we have been through certain things, so we can both go to a meeting and there is no problem”. Likewise, Marie (gestational mother of a 4-year-old son) reported, “And at school, I think that, from the start, I went to find the teacher and I said to her ‘There you go, I just wanted to explain a little situation to you’. So there you are, I was telling her, we explained to her, ‘Well there you go, Nora has a mother, a babou, two ladies, two persons, two ladies, two women! Who… well uh… who have a baby girl’. And then she changed schools”.

The choice of school could also be made according to the recommendations of family members and friends. In many cases, this allowed children to have bonds (i.e., with the children of those family members and friends) both inside and outside of the school, thus strengthening their relational networks. Some mothers also enrolled their child in the same school as their cousins, seeking to reduce their child’s feelings of loneliness. Indeed, for Adrianne (gestational mother of a 2-year-old daughter), the decision to put her child in the same school as her cousins constituted a relief, by telling herself that “the transition would perhaps be easier”. Finally, some mothers enrolled their child in a school where the teacher was known, in order to minimize the risk of parent–teacher disagreement.

However, the mothers were aware that they would not always have all of the answers to their child’s questions, and this was an additional source of stress. Mila (gestational mother of a 23-month-old daughter) explained that her greatest fear was an inability to find the right words and the right explanation to reassure her child at the right time: “They’re going to break my heart because I won’t get an answer. We can imagine, we cannot foresee all the questions, it is more for me these tests that await us”.

To prevent this situation, some mothers engaged in a process of documentation, information gathering, reading, and consulting specialists to obtain all of the necessary information. This strategy allowed them to form a global vision of their situation with hindsight, engaging in a search for meaning to arm themselves for difficult questions their child may ask about the family. As Hélène (gestational mother of 6-month-old twin sons) described, “Yes, books on same-sex parenting. We also have a book, The Mystery of Baby Seeds, for example. It will explain a bit of where they come from. Because we are very clear, also to say that this baby was not made just between us. There was a gift, from a man”. Likewise, Mila (gestational mother of a 23-month-old daughter) reported, “But we think it’s important because we say to ourselves that there shouldn’t be a lot of studying. We tried to learn, to read, to prepare for that and we really struggled to find documentation. I may have found five pounds at all broken. And these are always very concrete testimonies”.

One of the main questions that occupied mothers’ minds was the question of when they should tell their child about their origins. Was there an ideal age? Hélène (gestational mother of 6-month-old twin sons) presented a book that suggested that the ideal time was the child’s seventh birthday, as this was associated with the onset of the “age of reason”, or the formation of rational thought. However, as mentioned, most mothers had started talking to their child about the family situation, to some degree, on a day-to-day basis. It was not uncommon for the mothers to consult child psychologists for guidance and outside support in these efforts, as Adrianne (gestational mother of a 2-year-old daughter) noted: “Maybe we thought we would try to see a child psychologist to chat with him a little bit and see what he thought about it, which was better for the kid not to unsettle him”.

All of these ways of coping were aimed at countering heteronormativity and homophobic stigma. The adoption of these strategies seemed to reinforce the mothers’ sense of parenting competence and family balance, while helping to reduce their anxiety around the fact that their family deviated from the norm. Their central mission was to establish stability and safety in their environment, in order to convey security to their child and protect them from potential stigmatization from the outside world.

## 4. Discussion

The present study aimed at shedding light on the various stressors and coping strategies experienced by lesbian mothers through donor insemination in the Belgian context. The results indicated that, even if the lesbian couples went through the steps reported by Carter and McGoldrick [11] for explaining the transition to parenthood and analyzing the need to re-organize their family life and family space while transitioning from a dyad to a triad, the couples (similarly to other sexual minority couples) also underwent other particular stages. In particular, in this study, we were particularly attentive to second-order changes required to implement the transition from the stage of the couple formation to the stage of the family with young children, according to Carter and McGoldrick’s model [11]. This last stage involves three different tasks: the modification of the couple system to “create the space” for the child; the construction of parental roles; and the readjustment of relationships within extended families to include the roles of parents and grandparents. In addition, each lesbian couple had to consider how they were going to have a child. Since many options were available, each with its own consequences, this choice could take a lot of time and require various activities, including information seeking about procedures, contacting the relevant agencies, and planning medical interventions for donor insemination.

Moreover, couples also had to prepare themselves to deal with extrafamilial and intrafamilial questions, doubts, and a lack of understanding about their decision to have a child. The mothers perceived these further steps as stressors that they would not have to face if they were a “traditional” heterosexual parent family, and that had a significant detrimental effect on their mental health [15]. A further fundamental factor in the re-organization of family dynamics following the birth of the child was the mothers’ experiences of discrimination and stigmatization. As mentioned above, studies have reported that lesbian mothers are often stigmatized for their desire to have a child and create a family, even within their extended family [56,57,58].

From a clinical perspective, some of the most interesting results of the present study describe the mothers’ strategies and coping methods for managing stress and stigmatization. Although stigmatization and heteronormativity played significant roles in the mothers’ lives, they had various strategies to alleviate the stress that these caused. One such strategy was summarized by the phrase “to live happily, let’s stay hidden”, indicating that some mothers preferred to avoid public visibility as a lesbian mother family, in order to enjoy a quiet family life. Of note, numerous studies have investigated the themes of coming out and the public visibility of sexual minority people in various contexts, highlighting that hiding strategies significantly correlate with lower levels of mental health [59].

However, in the present study, “coming out” referred not to sexual identity, but to family life. This could be hidden in only a few scenarios, such as meeting with strangers; in other situations (e.g., in a hospital, at work, at school), it was not possible for the mothers to hide the nature of their parental relationships to the child. In these situations, many mothers reported episodes of discrimination and stigmatization, since they could not avoid others’ opinions of what a family should look like. Most of the mothers’ personal concerns about being discriminated against (or seeing their children discriminated against) were connected to these places, suggesting that further preventive actions should be undertaken to educate professionals in such contexts to better interact with non-traditional families. In this vein, future studies should investigate the consequences of strategies of social invisibility or visibility, especially on children’s self-esteem and internal representations of their family, over the long term.

A further interesting factor that the mothers perceived as a stressor concerned their child’s potential questions about their origins and genetic make-up, as well as why they have such a family and why they must face episodes of discrimination and marginalization. The mothers’ concerns underlined their need for better professional support; while they hoped to find answers to their child’s questions in articles and from practitioners, their efforts were often frustrated. Some of their doubts and worries associated with potential questions and stigmatization from either their children or their inner social circles may have been linked to internalized homophobia, as reported by previous studies. That is, they may have questioned themselves and their parenting skills due to an internalized idea that their sexual orientation made them less capable of raising and caring for their children [60]. 

Overall, in clinical work with lesbian mothers and their children, it may be helpful to consider that internalized homophobia may intertwine with, and inherently reflect, external negative views on family structures that differ from the traditional, cisgender, mother–father model. In this regard, news media have a central role as they may tend towards demonization, titillation, or constructions of their parenting model as a seduction-based breakdown of some traditional family ideal [61]. This external messaging can be, in partial ways, fundamentally educational for the audience, even if fallacious or discriminatory, by using representations of lesbian mother families to reinforce hetero-gendered norms [61,62]. It is unsurprising, therefore, that several studies on sexual minority parent families have emphasized a parents’ desire to be seen as “part of a broader diversity”, rather than specifically singled out for special attention as a sub-group (e.g., [62,63]). With this being the case, it cannot be excluded that in some ways the concerns for privacy or not sticking out expressed within this study may echo such sentiments. The clinical setting may be a safe context for examining and elaborating upon these concerns.

The current study presents some limitations. The first one refers to the sample size and the convenience nature of the sample which recruited members of LGBT+ associations and the authors’ acquaintances; these factors do not allow for generalizability of our results. Lastly, data were collected within a national sample and the mothers’ experiences and thoughts may not be representative of lesbian mothers living outside of Belgium or of those who went through donor insemination in this country later, with a different social and legal context.

## 5. Conclusions

The present study highlighted not only the stressors that lesbian mother families face, but also the resources and coping strategies that lesbian mothers deploy to deal with the obstacles they meet in their transition to parenthood. In doing so, this study contributes to advanced research into the psychology of sexual orientations, in the face of prejudices and stereotyping against sexual minority people that still pervade modern societies [64,65], as well as provides insight into the difficulties encountered by lesbian couples in countries in which only one mother has full legal recognition.

Of note, the present study focused on the experiences of lesbian mothers, and many of the mothers may have focused their attention on the situation they were experiencing at that time (i.e., post-birth). In this regard, future studies should seek to investigate the resources and strategies that lesbian mothers enact during the first steps of their transition to parenthood (i.e., from the moment of their decision to establish a family, through the pregnancy, and up to the birth). While previous studies have focused on the salient transitions that characterize intended lesbian mothers [10,15,18], there remains a lack of knowledge about the resilience and coping strategies of lesbian mothers in the face of obstacles and discrimination. Accordingly, the present study adds novel insight by identifying and discussing the types of stressors that lesbian mothers through donor insemination are likely to face, and the strategies they may rely on to counteract these stressors after the birth of their child and the constitution of their family.

## Data Availability

The data underlying this article cannot be shared publicly to protect the privacy of individuals that participated in the study.

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
