# Peer review of "Stress and Resilience Experiences during the Transition to Parenthood among Belgian Lesbian Mothers through Donor Insemination"

_ijerph, 2023, doi:10.3390/ijerph20042800_

Round 1

Reviewer 1 Report

This is an enjoyable paper! I note this study looked at a new cohort of parents in the LGBTIQA+ parenting research (specifically Belgian Lesbian Mothers, on whom I had not yet seen data). I recognise that it is well written, smooth to follow, and had some findings that both contribute to the existing international data on parents around public/privacy divides (see work by Jennifer Power and Henry von Doussa), and on the negativity on lesbian mothers' family constructions (see some of the points about the emphases on negative lesbian and bisexual mother constructions compared to more positive gay dad constructions in global news media in Trent Mann and Tiffany Jones'  (2022). Including LGBT Parented Families in Schools: Research to Inform Policy and Practice. Routledge: London)...

Here I feel the article could note in the discussion, if such literature were to be included in the literature review, that the final finding around negative views on family structures may go a little beyond internalised homophobia. I am concerned at this time it is too simply written off as such. In many countries the news media on the family compositions of lesbian mothers tends towards demonisation, titillation, or constructions of their parenting model as a seduction-based break down of some 'original' family ideal. This external messaging likely impacts mothers, their children and their extended families' constructions of their family model as news media can be 'educative' even if fallacious or discriminatory. There can be some misogyny in such constructions of mothers, including celebrity lesbian and bisexual mothers, and a tendency to see them as rebelling from normative roles. By comparison global news media has often emphasised the financial stability and wealth of gay dads, particularly celebrity gay dads. This seems an unfair and artificial construction likely to impact how the lesbian mothers are read and read themselves.

It is also noteworthy that many studies on LGBT parents/guardians have emphasised a desire to be seen as 'part of a broader diversity' rather than specifically singled out for special attention as a sub-group (see the Mann and Jones 2022 book, the work of Goldberg across several studies, Carlile et al. etc.). In some ways the concerns for privacy or not sticking out expressed within this study may echo such sentiments, or be a local permeation of this wider theme?

Well done overall on some interesting work. It was well presented and could perhaps unpack some of these findings, echoing the wider research, just a little more in a line or two here and there.

Reviewer 2 Report

Review Report

I greatly appreciated the opportunity to read the manuscript “A Qualitative Investigation of Parenting, Stress, and Resilience Experiences among Belgian Lesbian Mothers through Donor Insemination”. In this study, 16 Belgian lesbian-women mothers were interviewed about their parenting aspirations, the impact of stigma on their stress, and their strategies of resilience. Data were analysed using thematic analysis and four themes were identified. Mothers reported some sources of stress and developed resilience strategies concerning the experiences of donor conception, the legal obstacles encountered, coming out, the non-gestational mother’s role, and coparenting.

This study contributes to the understanding of the lived experience of motherhood by lesbian-women mothers couples providing some advancement of the current knowledge. The manuscript is clear, relevant for the field and presented in a well-structured manner. The cited studies are relevant and recent. The conclusions are consistent with the evidence and arguments presented.

I would like to give some minor suggestions to improve the connections between the conceptual and empirical grounding of the paper and the interpretation of results, including their limitations.

Below, I will provide my comments in detail.

Abstract section: It is unclear what the mean and standard deviation refer to. This information can be clearly stated in the Method section and removed from the abstract.

p. 2: I would suggest consulting also the ILGA 2022 index in addition to the CCNE index.

p. 2 (lines 69-70): The authors introduced an interesting topic that intersects with that of donor insemination: the adoption process of non-gestational mother. The authors stated that: “However, the legal situation is such that, if a female parental couple is legally married, the non-gestational mother must adopt the child in order for her to be legally recognized as a mother.[....] any reinforcement of this model puts significant pressure on sexual minority couples and makes their transition to parenthood unequal to that of heterosexual couples.”Authors could capitalize the unique contribution of the study to the vast existent literature on this topic adding some specific information about such an adoption process in Belgium. How is it structured? Are there differences in the aspiration and perceptions of this adoption process and a "full" adoption process in which both mothers are non-gestational parents? Furthermore, I understand from what the authors reported that a form of discrimination is applied in Belgium. If so, perhaps it should be explicated - providing a bibliographical reference - that in infertile married different-gender couples using insemination, the adoption process of non biological parent is simpler/automatic, compared to same-gender parents.

pp. 2 and 3 (lines 94-99): The results of Bos and colleagues' study are reported in the lines 94-99, but also a few lines above. I would suggest reorganizing that paragraph.

p. 3: Among the critical points presented, it might also be relevant to mention the lesbian women's difficulty in confronting the heterosexist norms (sometimes internalized) according to which a lesbian woman's identity and motherhood were incompatible. This norm stem from a heteronormative image of family and a gendered perception of women as nurturers, caregivers, and protectors of children (see also in Allen, K. R., & Goldberg, A. E. (2020). Lesbian women disrupting gendered, heteronormative discourses of motherhood, marriage, and divorce. Journal of lesbian studies, 24(1), 12-24.).

p. 5: In the Procedure section, some more information could be added about how mothers were recruited, and that it was a convenience sample. Also, it might be helpful to understand if there is any indication of the adoption pathway of the non-gestational mother: Had all the non-gestational mothers in the couples already adopted their children?

p. 6: The authors reported that: “The third theme refers to the mothers’ lack of legal recognition of parental rights prior to the change in law recognizing stepchild adoption. In the previous legal context, the non-gestational mother, when unmarried and not the adoptive parent, had no rights and no obligations towards the child, leading to fear over a loss of parental rights. As reported by Elodie (gestational mother of a 1-year-old daughter): Alice (Elodie’s non-gestational coparent) had absolutely no rights at all. From the moment the child was born, she had nothing more to say. It was quite complicated for us, and it worried me a bit. There was a long journey for my partner to legally become a mother to her daughter.” From this paragraph, I understand that in the last year (age of the child) there has been a major legal change in Belgium concerning adoption but that this adoption process continues to be difficult for same-gender couples. This adoption process and the legal change should be made more explicit in the theoretical part of the paper.

p. 9: Consider whether best to use the term sexual orientation or sexual identity following APA standards: https://apastyle.apa.org/style-grammar-guidelines/bias-free-language/sexual-orientation#:~:text=Use%20the%20term%20%E2%80%9Csexual%20orientation,itself%20is%20not%20a%20choice.

Discussion section: In the first lines of the paper and then in the discussions, Carter and McGoldrick's family life cycle model is mentioned. It might be helpful to add some information about the steps of this model and how they guided the analyses in this study.

A limitation section is missing.

Reviewer 3 Report

I was pleased to review the manuscript “A Qualitative Investigation of Parenting, Stress, and Resilience Experiences among Belgian Lesbian Mothers through Donor Insemination” as it deals with a timely topic and provides relevant insights for clinical work with lesbian mothers with donor-conceived children.

The manuscript is well written and theoretically grounded; methodologically, the use of reflexive thematic analysis is appropriate.  I would be glad to recommend it for publication after a few of very minor revisions which I hope the authors might find useful to improve their manuscript. 

- In my opinion, the title, in its current form does not vehicle the manuscript focus on a straightforward manner. I would suggest rewording it stressing the transition to parenthood. As an example, I might suggest:“Stress and Resilience Experiences during the Transition to Parenthood among Belgian Lesbian Mothers through Donor Insemination”;

-  In the paragraph 1.1 “Stigma Generating Stress for Lesbian Mother Families”, I would suggest to elaborate the stereotypes and prejudice that lesbian mother families face in modern societies. The authors write that “ […] they may experience rejection in the form of regular irritating and intrusive questions regarding their 120 family life, and they may also doubt their own parenting competency.” I agree with them. The following references might help the authors to elaborate more on this: Di Battista et al. (2022). Attitudes toward “Non-Traditional” Mothers: Examining the Antecedents of Mothers’ Competence Perceptions. Social Sciences, 11(11), 495.https://doi.org/10.3390/socsci11110495Di Battista et al.,  (2023). Attitudes Toward Heterosexual and Lesbian Stepmothers: An Experimental Test in the Italian Context. Journal of Family Issueshttps://doi.org/10.1177/0192513X2211509

-  I would suggest the authors adding a short sparagraph at the end of the Discussion section (or it can be included in the discussions too) about the limitations of the current research. Here, the authors might write something about the sample size, the generalizability of the results based on sample’s characteristics, the qualitative nature of their data, etc…

I appreciated the Conclusions of the authors. I would suggest authors to emphasize more that the present study adds novel insights and contributes to advance research into the psychology of sexual orientations and gender identities, in the face of LGBTQ prejudices and stereotyping that still pervade modern societies. You may refers to the following work to elabore more on this: Salvati, M., & Koc, Y. (2022). Advancing research into the social psychology of sexual orientations and gender identities: Current research and future directions. European Journal of Social Psychology, 52(2), 225-232. https://doi.org/10.1002/ejsp.2875; Salvati, et al., (2020). Introduction to the special issue: Sexual prejudice and stereotyping in modern societies. Psicologia sociale, 15(1), 5-14.10.1482/96291
